# Outlier Detection and Robust PCA Using a Convex Measure of Innovation

**Mostafa Rahmani and Ping Li**
Cognitive Computing Lab
Baidu Research
10900 NE 8th St. Bellevue, WA 98004, USA
`{mostafarahmani,liping11}@baidu.com`

## Abstract

This paper presents a provable and strong algorithm, termed Innovation Search (iSearch), to robust Principal Component Analysis (PCA) and outlier detection. An outlier by definition is a data point which does not participate in forming a low dimensional structure with a large number of data points in the data. In other words, an outlier carries some innovation with respect to most of the other data points. iSearch ranks the data points based on their values of innovation. A convex optimization problem is proposed whose optimal value is used as our measure of innovation. We derive analytical performance guarantees for the proposed robust PCA method under different models for the distribution of the outliers including randomly distributed outliers, clustered outliers, and linearly dependent outliers. Moreover, it is shown that iSearch provably recovers the span of the inliers when the inliers lie in a union of subspaces. In the challenging scenarios in which the outliers are close to each other or they are close to the span of the inliers, iSearch is shown to outperform most of the existing methods.

## 1 Introduction

Outlier detection is an important research problem in unsupervised machine learning. Outliers are associated with important rare events such as malignant tissues [14], the failures of a system [10, 12, 31], web attacks [16], and misclassified data points [9, 27]. In this paper, the proposed outlier detection method is introduced as a robust Principal Component Analysis (PCA) algorithm, i.e., the inliers lie in a low dimensional subspace. In the literature of robust PCA, two main models for the data corruption are considered: the element-wise model and the column-wise model. These two models are corresponding to two different robust PCA problems. In the element-wise model, it is assumed that a small subset of the elements of the data matrix are corrupted and the support of the corrupted elements is random. This problem is known as the low rank plus sparse matrix decomposition problem [1, 3, 4, 23, 24]. In the column-wise model, a subset of the columns of the data are affected by the data corruption [5, 7, 8, 11, 17, 20, 25, 26, 36–39]. Section 2 provides a review of the robust (to column-wise corruption) PCA methods. This paper focuses on the column-wise model, i.e., we assume that the given data follows Data Model 1.

**Data Model 1.** *The data matrix $\mathbf{D} \in \mathbb{R}^{M_1 \times M_2}$ can be expressed as $\mathbf{D} = [\mathbf{B} \; (\mathbf{A} + \mathbf{N})] \, \mathbf{T}$, where $\mathbf{A} \in \mathbb{R}^{m \times n_i}$, $\mathbf{B} \in \mathbb{R}^{m \times n_o}$, $\mathbf{T}$ is an arbitrary permutation matrix, and $[\mathbf{B} \; (\mathbf{A} + \mathbf{N})]$ represents the concatenation of $\mathbf{B}$ and $(\mathbf{A} + \mathbf{N})$. The columns of $\mathbf{A}$ lie in an $r$-dimensional subspace $\mathcal{U}$. The columns of $\mathbf{B}$ do not lie entirely in $\mathcal{U}$, i.e., the $n_i$ columns of $\mathbf{A}$ are the inliers and the $n_o$ columns of $\mathbf{B}$ are the outliers. The matrix $\mathbf{N}$ represents additive noise. The orthonormal matrix $\mathbf{U} \in \mathbb{R}^{M_1 \times r}$ is a basis for $\mathcal{U}$. Evidently, $M_2 = n_i + n_o$.*

In the robust PCA problem, the main task is to recover $\mathcal{U}$. Clearly, if $\mathcal{U}$ is estimated accurately, the outliers can be located using a simple subspace projection [22].

**Summary of Contributions:** The main contributions can be summarized as follows.

- The proposed approach introduces a new idea to the robust PCA problem. iSearch uses a convex optimization problem to measure the Innovation of the data points. It is shown that iSearch mostly outperforms the exiting methods in handling close outliers and noisy data.

- To the best of our knowledge, the proposed approach and the CoP method presented in [27] are the only robust PCA methods which are supported with analytical performance guarantees under different models for the distributions of the outliers including the randomly distributed outliers, the clustered outliers, and the linearly dependent outliers.

- In addition to considering different models for the distribution of the outliers, we provide analytical performance guarantees under different models for the distributions of the inliers too. The presumed models include the union of subspaces and the uniformly at random distribution on $\mathcal{U} \cap \mathbb{S}^{M_1-1}$ where $\mathbb{S}^{M_1-1}$ denotes the unit $\ell_2$-norm sphere in $\mathbb{R}^{M_1}$.

**Notation:** Given a matrix $\mathbf{A}$, $\|\mathbf{A}\|$ denotes its spectral norm. For a vector $\mathbf{a}$, $\|\mathbf{a}\|_p$ denotes its $\ell_p$-norm and $\mathbf{a}(i)$ its $i^{\text{th}}$ element. Given two matrices $\mathbf{A}_1$ and $\mathbf{A}_2$ with an equal number of rows, the matrix $\mathbf{A}_3 = [\mathbf{A}_1 \ \mathbf{A}_2]$ is the matrix formed by concatenating their columns. For a matrix $\mathbf{A}$, $\mathbf{a}_i$ denotes its $i^{\text{th}}$ column. The subspace $\mathcal{U}^\perp$ is the complement of $\mathcal{U}$. The cardinality of set $\mathcal{I}$ is defined as $|\mathcal{I}|$. Also, for any positive integer n, the index set $\{1, ..., n\}$ is denoted $[n]$. The coherence between vector $\mathbf{a}$ and subspace $\mathcal{H}$ with orthonormal basis $\mathbf{H}$ is defined as $\|\mathbf{a}^T \mathbf{H}\|_2$.

# 2 Related Work

In this section, we briefly review some of the related works. We refer readers to [18, 27] for a more comprehensive review on the topic. One of the early approaches to robust PCA was to replace the Frobenius norm in the cost function of PCA with $\ell_1$-norm because $\ell_1$-norm were shown to be robust to the presence of the outliers [2,15]. The method proposed in [6] leveraged the column-wise structure of the corruption matrix and replaced the $\ell_1$-norm minimization problem with an $\ell_{1,2}$-norm minimization problem. In [19] and [39], the optimization problem used in [6] was relaxed to a convex optimization problem and it was proved that under some sufficient conditions the optimal point is a projection matrix which spans $\mathcal{U}$. In [34], a provable outlier rejection method was presented. However, [34] assumed that the outliers are randomly distributed on $\mathbb{S}^{S-1}$ and the inliers are distributed randomly on $\mathcal{U} \cap \mathbb{S}^{M_1-1}$. In [36], a convex optimization problem was proposed which decomposes the data into a low rank component and a column sparse component. The approach presented in [36] is provable but it requires $n_o$ to be significantly smaller than $n_i$. In [32], it was assumed that the outliers are randomly distributed on $\mathbb{S}^{M_1-1}$ and a small number of them are not linearly dependent. The method presented in [32] detects a data point as an outlier if it does not have a sparse representation with respect to the other data points.

**Connection and Contrast to Coherence Pursuit:** In [27], Coherence Pursuit (CoP) was proposed as a provable robust PCA method. CoP computes the Coherence Values for all the data points to rank the data points. The Coherence value corresponding to data column $\mathbf{d}$ is a measure of resemblance between $\mathbf{d}$ and the rest of the data columns. CoP uses the inner product between $\mathbf{d}$ and the rest of the data points to measure the resemblance between $\mathbf{d}$ and the rest of data. In sharp contrast, iSearch finds an optimal direction corresponding to each data column. The optimal direction corresponding to data column $\mathbf{d}$ is used to measure the innovation of $\mathbf{d}$ with respect to the rest of the data columns. We show through theoretical studies and numerical experiments that finding the optimal directions makes iSearch significantly stronger than CoP in detecting outliers which carry weak innovation.

**Connection and Contrast to Innovation Pursuit:** In [28, 29], Innovation Pursuit was proposed as a new subspace clustering method. The optimization problem proposed in [28] finds a direction in the span of the data such that it is orthogonal to the maximum number of data points. We present a new discovery about the applications of Innovation Pursuit. It is shown that the idea of innovation search can be used to design a strong outlier detection algorithm. iSearch uses an optimization problem similar to the linear optimization problem used in [28] to measure the innovation of the data points.

---

**Algorithm 1** Subspace Recovery Using iSearch

---

**1. Data Preprocessing.** The input is data matrix $\mathbf{D} \in \mathbb{R}^{M_1 \times M_2}$.

**1.1** Define $\mathbf{Q} \in \mathbb{R}^{M_1 \times r_d}$ as the matrix of first $r_d$ left singular vectors of $\mathbf{D}$ where $r_d$ is the number of non-zero singular values. Set $\mathbf{D} = \mathbf{Q}^T\mathbf{D}$. If dimensionality reduction is not required, skip this step.

**1.2** Normalize the $\ell_2$-norm of the columns of $\mathbf{D}$, i.e., set $\mathbf{d}_i$ equal to $\mathbf{d}_i/\|\mathbf{d}_i\|_2$ for all $1 \leq i \leq M_2$.

**2. Direction Search.** Define $\mathbf{C}^* \in \mathbb{R}^{r_d \times M_2}$ such that $\mathbf{c}_i^* \in \mathbb{R}^{r_d \times 1}$ is the optimal point of

$$\min_{\mathbf{c}} \|\mathbf{c}^T\mathbf{D}\|_1 \quad \text{subject to} \quad \mathbf{c}^T\mathbf{d}_i = 1$$

or define $\mathbf{C}^* \in \mathbb{R}^{r_d \times M_2}$ as the optimal point of

$$\min_{\mathbf{C}} \|(\mathbf{C}^T\mathbf{D})^T\|_1 \quad \text{subject to} \quad \text{diag}(\mathbf{C}^T\mathbf{D}) = \mathbf{1} \,. \tag{1}$$

**3. Computing the Innovation Values.** Define vector $\mathbf{x} \in \mathbb{R}^{M_2 \times 1}$ such that $\mathbf{x}(i) = 1/\|\mathbf{D}^T\mathbf{c}_i^*\|_1$.

**4. Building Basis.** Construct matrix $\mathbf{Y}$ from the columns of $\mathbf{D}$ corresponding to the smallest elements of $\mathbf{x}$ such that they span an $r$-dimensional subspace.

**Output:** The column-space of $\mathbf{Y}$ is the identified subspace.

---

# 3 Proposed Approach

Algorithm 1 presents the proposed method along with the definition of the used symbols. iSearch consists of 4 steps. In the next subsections, Step 2 and Step 4 are discussed. In this paper, we use an ADMM solver to solve (1). The computation complexity of the solver is $\mathcal{O}(\max(M_1 M_2^2, M_1^2 M_2))$. If PCA is used in the prepossessing step to reduce the dimensionality of the data to $r_d$, the computation complexity of the solver is $\mathcal{O}(\max(r_d M_2^2, r_d^2 M_2))$ [1].

## 3.1 An Illustrative Example for Innovation Value

We use a synthetic numerical example to explain the idea behind the proposed approach. Suppose $\mathbf{D} \in \mathbb{R}^{20 \times 250}$, $n_i = 200$, $n_o = 50$, and $r = 3$. Assume that $\mathbf{D}$ follows Assumption 1.

**Assumption 1.** *The columns of $\mathbf{A}$ are drawn uniformly at random from $\mathcal{U} \cap \mathbb{S}^{M_1-1}$. The columns of $\mathbf{B}$ are drawn uniformly at random from $\mathbb{S}^{M_1-1}$. To simplify the exposition and notation, it is assumed without loss of generality that $\mathbf{T}$ in Data Model 1 is the identity matrix, i.e, $\mathbf{D} = [\mathbf{B} \ \mathbf{A}]$.*

Suppose $\mathbf{d}$ is a column of $\mathbf{D}$, define $\mathbf{c}^*$ as the optimal point of

$$\min_{\mathbf{c}} \|\mathbf{c}^T\mathbf{D}\|_1 \quad \text{subject to} \quad \mathbf{c}^T\mathbf{d} = 1 \,, \tag{2}$$

and define the Innovation Value corresponding to $\mathbf{d}$ as $1/\|\mathbf{D}^T\mathbf{c}^*\|_1$. The main idea of iSearch is that $\mathbf{c}^*$ has two completely different behaviours with respect to $\mathcal{U}$ (when $\mathbf{d}$ is an outlier and when $\mathbf{d}$ is an inlier). Suppose $\mathbf{d}$ is an outlier. The optimization problem (2) searches for a direction whose projection on $\mathbf{d}$ is non-zero and it has the minimum projection on the rest of the data points. As $\mathbf{d}$ is an outlier, $\mathbf{d}$ has a non-zero projection on $\mathcal{U}^\perp$. In addition, as $n_i$ is large, (2) searches for a direction in the ambient whose projection on $\mathcal{U}$ is as weak as possible. Thus, $\mathbf{c}^*$ lies in $\mathcal{U}^\perp$ or it is close to $\mathcal{U}^\perp$.

The left plot of Figure 1 shows $\mathbf{D}^T\mathbf{c}^*$ when $\mathbf{d}$ is an outlier. In this case, $\mathbf{c}^*$ is orthogonal to all the inliers. Accordingly, when $\mathbf{d}$ is an outliers, $\|\mathbf{D}^T\mathbf{c}^*\|_1$ is approximately equal to $\|\mathbf{B}^T\mathbf{c}^*\|_1$. On the other hand, when $\mathbf{d}$ is an inlier, the linear constraint strongly discourages $\mathbf{c}^*$ to lie in $\mathcal{U}^\perp$ or to be close to $\mathcal{U}^\perp$. Inliers lie in a low dimensional subspace and mostly they are close to each other. Since $\mathbf{c}^*$ has a strong projection on $\mathbf{d}$, it has strong projections on many of the inliers. Accordingly, the value of $\|\mathbf{A}^T\mathbf{c}^*\|_1$ is much larger when $\mathbf{d}$ is an inlier. Therefore, the Innovation Value corresponding to an inlier is smaller than the Innovation Value corresponding to an outlier because $\|\mathbf{A}^T\mathbf{c}^*\|_1$ is much larger when $\mathbf{d}$ is an inliers. Figure 1 compares the vector $\mathbf{D}^T\mathbf{c}^*$ when $\mathbf{d}$ is an outliers with the same vector when $\mathbf{d}$ is an inlier. In addition, it shows the vector of Innovation Values (right plot). One can observe that the Innovation Values make the outliers clearly distinguishable.

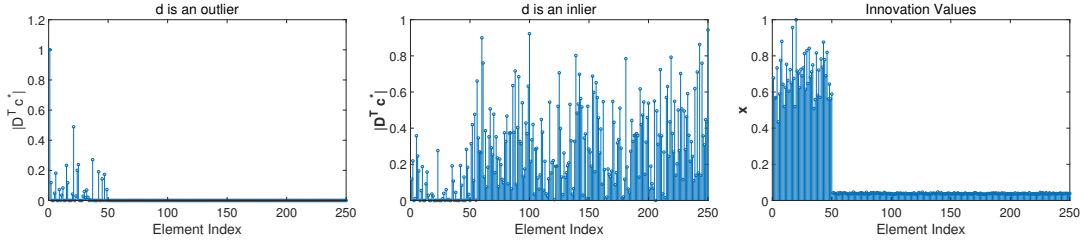

Figure 1: The first 50 columns are outliers. The left panel shows vector $\mathbf{D}^T\mathbf{c}^*$ when $\mathbf{d}$ is an outlier. The middle panel depicts $\mathbf{D}^T\mathbf{c}^*$ when $\mathbf{d}$ is an inlier. The right panel shows the Innovation Values corresponding to all the data points (vector $\mathbf{x}$ was defined in Algorithm 1).

## 3.2 Building the Basis Matrix

The data points corresponding to the least Innovation Values are used to construct the basis matrix $\mathbf{Y}$. If the data follows Assumption 1, the $r$ data points corresponding to the $r$ smallest Innovation Values span $\mathcal{U}$ with overwhelming probability [35]. In practise, the algorithm should continue adding new columns to $\mathbf{Y}$ until the columns of $\mathbf{Y}$ spans an $r$-dimensional subspace. This approach requires to check the singular values of $\mathbf{Y}$ several times. We propose two techniques to avoid this extra steps. The first approach is based on the side information that we mostly have about the data. In many applications, we can have an upper-bound on $n_o$ because outliers are mostly associated with rare events. If we know that the number of outliers is less than $y$ percent of the data, matrix $\mathbf{Y}$ can be constructed using $(1-y)$ percent of the data columns which are corresponding to the least Innovation Values. The second approach is the adaptive column sampling method proposed in [27]. The adaptive column sampling method avoids sampling redundant columns.

## 4 Theoretical Studies

In this section, we analyze the performance of the proposed approach with three different models for the distribution of the outliers: unstructured outliers, clustered outliers, and linearly dependent outliers. Moreover, we analyze iSearch with two different models for the distribution of the inliers. These models include the union of subspaces and uniformly at random distribution on $\mathcal{U} \cap \mathbb{S}^{M_1-1}$. Due to space limitation, in this paper we do not include theoretical guarantees with noisy data and we refer the reader for the analysis of iSearch with noisy data to [30]. In Section 5, it is shown with real and synthetic data that iSearch accurately detects the outliers even in the low signal to noise ratio cases and it mostly outperforms the existing approaches when the data is noisy. The theoretical results are followed by short discussions which highlight the important aspects of the theorems. The proofs of the presented theorems are available in an extended version of this work [30].

### 4.1 Randomly Distributed Outliers

In this section, it is assumed that $\mathbf{D}$ follows Assumption 1. In order to guarantee the performance of the proposed approach, it is enough to show that the Innovation Values corresponding to the outliers are greater than the Innovation Values corresponding to the inliers. In other word, it suffices to show

$$\max\left(\{1/\|\mathbf{D}^T\mathbf{c}_i^*\|_1\}_{i=n_o+1}^{M_2}\right) < \min\left(\{1/\|\mathbf{D}^T\mathbf{c}_j^*\|_1\}_{j=1}^{n_o}\right) . \tag{3}$$

Before we state the theorem, let us provide the following definitions and remarks.

**Definition 1.** *Define* $\mathbf{c}_j^* = \underset{\mathbf{d}_j^T\mathbf{c}=1}{\arg\min} \ \|\mathbf{c}^T\mathbf{D}\|_1$. *In addition, define* $\chi = \max\left(\{\|\mathbf{c}_j^*\|_2\}_{i=1}^{n_o}\right)$, *and* $n_z' = \max\left(\{|\mathcal{I}_0^i|\}_{i=1}^{n_o}\right)$ *where* $\mathcal{I}_0^i = \{i \in [n_o] : \mathbf{c}_i^{*T}\mathbf{b}_i = 0\}$ *and* $\mathbf{b}_i$ *is the* $i^{th}$ *column of* $\mathbf{B}$. *The value* $|\mathcal{I}_0^i|$ *is the number of outliers which are orthogonal to* $\mathbf{c}_i^*$.

**Remark 1.** *In Assumption 1, the outliers are randomly distributed. Thus, if* $n_o$ *is significantly larger than* $M_1$, $n_z'$ *is significantly smaller than* $n_o$ *with overwhelming probability.*

**Theorem 1.** *Suppose $\mathbf{D}$ follows Assumption 1 and define $\mathcal{A} = \sqrt{\frac{1}{2\pi}} \frac{n_i}{\sqrt{r}} - \sqrt{n_i} - \sqrt{\frac{n_i \log \frac{1}{\delta}}{2r-2}}$. If*

$$\mathcal{A} > \left[ \frac{n_o}{M_1} + 2\sqrt{\frac{n_o}{M_1}} + \sqrt{\frac{2n_o \log 1/\delta}{(M_1-1)M_1}} + \sqrt{\frac{n_o c_\delta'' \log n_o/\delta}{M_1^2}} + \right.$$

$$\left. n_z' \sqrt{\frac{c_\delta''}{M_1^2}} + \sqrt{\left(\frac{n_o}{M_1^2} + \frac{\eta_\delta}{M_1}\right)\log n_o/\delta} \right] \sqrt{\frac{4M_1 c_\delta}{M_1 - c_\delta r}} \quad and \tag{4}$$

$$\mathcal{A} > \max\left( \chi \frac{n_o}{\sqrt{M_1}} + 2\sqrt{n_o}(1+\sqrt{\chi}) + 2\sqrt{\frac{2\chi n_o \log \frac{1}{\delta}}{M-1}}, 2n_z'\sqrt{\frac{c_\delta r}{M_1}} + 2\sqrt{\frac{n_o c_\delta r \log n_o/\delta}{M_1}} \right),$$

*then (3) holds and $\mathcal{U}$ is recovered exactly with probability at least $1 - 7\delta$ where $\sqrt{c_\delta} = 3\max\left(1, \sqrt{\frac{8M_1\pi}{(M_1-1)r}}, \sqrt{\frac{8M_1 \log n_o/\delta}{(M_1-1)r}}\right)$, $\sqrt{c_\delta''} = 3\max\left(1, \sqrt{\frac{8M_1\pi}{M_1-1}}, \sqrt{\frac{16M_1 \log n_o/\delta}{M_1-1}}\right)$, and $\eta_\delta = \max\left(\frac{4}{3}\log\frac{2M_1}{\delta}, \sqrt{4\frac{n_o}{M_1}\log\frac{2M_1}{\delta}}\right)$.*

Theorem 1 shows that as long as $n_i/r$ is sufficiently larger than $n_o/M_1$, the proposed approach is guaranteed to detect the randomly distributed outliers exactly. It is important to note that in the sufficient conditions $n_i$ is scaled with $1/r$ but $n_o$ is scaled with $1/M_1$. It shows that if $r$ is sufficiently smaller than $M_1$, iSearch provably detects the unstructured outliers even if $n_o$ is much larger than $n_i$. The numerical experiments presented in Section 5 confirms this feature of iSearch and they show that if the outliers are unstructured, iSearch can yield exact recovery even if $n_o > 100\,n_i$. It is important to note that when the outliers are structured, by the definition of outlier, $n_o$ cannot be larger than $n_i$.

## 4.2 Structured Outliers

In this section, we analyze the proposed approach with structured outliers. In contrast to the unstructured outliers, structured outliers can form a low dimensional structure different from the structure of the majority of the data points. Structured outliers are associated with important rare events such as malignant tissues [14] or web attacks [16]. In this section, we assume that the outliers form a cluster outside of $\mathcal{U}$. The following assumption specifies the presumed model for the distribution of the structured outliers.

**Assumption 2.** *A column of $\mathbf{B}$ is formed as $\mathbf{b}_i = \frac{1}{\sqrt{1+\eta^2}}(\mathbf{q} + \eta\mathbf{v}_i)$. The unit $\ell_2$-norm vector $\mathbf{q}$ does not lie in $\mathcal{U}$, $\{\mathbf{v}_i\}_{i=1}^{n_o}$ are drawn uniformly at random from $\mathbb{S}^{M_1-1}$, and $\eta$ is a positive number.*

According to Assumption 2, the outliers cluster around vector $\mathbf{q}$ where $\mathbf{q} \notin \mathcal{U}$. In Algorithm 1, if the dimensionality reduction step is performed, the direction search optimization problem is applied to $\mathbf{Q}^T \mathbf{D}$. Thus, (2) is equivalent to

$$\min_{\mathbf{c}} \|\mathbf{c}^T\mathbf{D}\|_1 \quad \text{subject to} \quad \mathbf{c}^T\mathbf{d} = 1 \quad \text{and} \quad \mathbf{c} \in \mathcal{Q}, \tag{5}$$

where $\mathbf{c} \in \mathbb{R}^{M_1 \times 1}$ and $\mathbf{D} \in \mathbb{R}^{M_1 \times M_2}$. The subspace $\mathcal{Q}$ is the column-space of $\mathbf{D}$. In this section, we are interested in studying the performance of iSearch in identifying tightly clustered outliers because some of the existing outlier detection algorithms fail if the outliers form a tight cluster. For instance, the thresholding based method [13] and the sparse representation based algorithm [32] fail when the outliers are close to each other. Therefore, we assume that the span of $\mathbf{Q}$ is approximately equal to the column-space of $[\mathbf{U} \ \mathbf{q}]$. The following Theorem shows that even if the outliers are close to each other, iSearch successfully identifies the outliers provided that $n_i/\sqrt{r}$ is sufficiently larger than $n_o$.

**Theorem 2.** *Suppose the distribution of the inliers/outliers follows Assumption-1/Assumption-2. Assume that $\mathcal{Q}$ is equal to the column-space of $[\mathbf{U} \ \mathbf{q}]$. Define $\mathbf{q}^\perp = \frac{(\mathbf{I}-\mathbf{U}\mathbf{U}^T)\mathbf{q}}{\|(\mathbf{I}-\mathbf{U}\mathbf{U}^T)\mathbf{q}\|_2}$, define $\beta = \max\left(\{1/|\mathbf{d}_i^T\mathbf{q}^\perp| \ : \mathbf{d}_i \in \mathbf{B}\}\right)$, define $\mathbf{c}_i^*$ as the optimal point of (5) with $\mathbf{d} = \mathbf{d}_i$, and assume that*

$\eta < |\mathbf{q}^T\mathbf{q}^\perp|$. *In addition, define* $\mathcal{A} = \frac{\sqrt{1+\eta^2}}{2\beta}\left(\sqrt{\frac{2}{\pi}}\frac{n_i}{\sqrt{r}} - 2\sqrt{n_i} - \sqrt{\frac{2n_i\log\frac{1}{\delta}}{r-1}}\right)$. *If*

$$\mathcal{A} > n_o\|\mathbf{U}^T\mathbf{q}\|_2 + \eta\sqrt{\frac{n_o r c_\delta \log n_o/\delta}{M_1}}\,,$$

$$\mathcal{A} > n_o|\mathbf{q}^T\mathbf{q}^\perp| + n_o\eta\sqrt{\frac{c_\delta'' \log n_o/\delta}{M_1}}\,,$$

(6)

*then (3) holds and* $\mathcal{U}$ *is recovered exactly with probability at least* $1 - 5\delta$.

In contrast to (4), in (6) $n_o$ is not scaled with $1/\sqrt{M_1}$. Theorem 2 shows that in contrast to the unstructured outliers, the number of the structured outliers should be sufficiently smaller than the number of the inliers for the small values of $\eta$. This is consistent with our intuition regarding the detection of structured outliers. If the columns of $\mathbf{B}$ are highly structured and most of the data points are outliers, it violates the definition of outlier to label the columns of $\mathbf{B}$ as outliers.

The presence of parameter $\beta$ emphasizes that the closer the outliers are to $\mathcal{U}$, the harder it is to distinguish them. In Section 5, it is shown that iSearch significantly outperforms the existing methods when the outliers are close to $\mathcal{U}$. The main reason is that even if an outlier is close to $\mathcal{U}$, its corresponding optimal direction obtained by (2) is highly incoherent with $\mathbf{U}$. Therefore, its corresponding optimal direction is incoherent with the inliers.

When the outliers are very close to the span of the inliers, the norm of $\mathbf{c}^*$ should be large to satisfy the linear constraint of (2) because $\mathbf{c}^*$ is orthogonal or nearly orthogonal to $\mathcal{U}$. Accordingly, in the applications in which the outliers are highly coherent with $\mathcal{U}$, the $\ell_2$-norm of $\mathbf{c}^*$ should be normalized before computing the Innovation Values.

## 4.3 Linearly Dependent Outliers

In some applications, the outliers are linearly dependent. For instance, in [9], it was shown that a robust PCA algorithm can be used to reduce the clustering error of a subspace segmentation method. In this application, a small subset of the outliers can be linearly dependent. This section focuses on detecting linearly dependent outliers. The following assumption specifies the presumed model for matrix $\mathbf{B}$ and Theorem 3 provides the guarantees.

**Assumption 3.** *Define subspace* $\mathcal{U}_o$ *with dimension* $r_o$ *such that* $\mathcal{U}_o \notin \mathcal{U}$ *and* $\mathcal{U} \notin \mathcal{U}_o$. *The outliers are randomly distributed on* $\mathbb{S}^{M_1-1} \cap \mathcal{U}_o$. *The orthonormal matrix* $\mathbf{U}_o \in \mathbb{R}^{M_1 \times r_o}$ *is a basis for* $\mathcal{U}_o$.

**Theorem 3.** *Suppose the distribution of the inliers/outliers follows Assumption-1/Assumption-3. Define* $\mathcal{A} = \sqrt{\frac{2}{\pi}}\frac{n_i}{\sqrt{r}} - 2\sqrt{n_i} - \sqrt{\frac{2n_i\log\frac{1}{\delta}}{r-1}}$. *If*

$$\mathcal{A} > 2n_z'\|\mathbf{U}^T\mathbf{U}_o\| + 2\|\mathbf{U}^T\mathbf{U}_o\|\sqrt{n_o \log n_o/\delta}\,,$$

$$\mathcal{A} > \frac{2\|\mathbf{U}^T\mathbf{U}_o\|}{\xi}\left(\frac{n_o}{\sqrt{r_o}} + 2\sqrt{n_o} + \sqrt{\frac{2n_o\log\frac{1}{\delta}}{r_o-1}} + 2\sqrt{\left(\frac{n_o}{r_o} + \eta_\delta'\right)\log\frac{n_o}{\delta}} + n_z'\right),$$

$$\mathcal{A} > \left(\frac{\chi n_o}{\sqrt{r_o}} + 2\sqrt{\chi n_o} + \sqrt{\chi\frac{2n_o\log\frac{1}{\delta}}{r_o-1}}\right)\|\mathbf{U}_o^T\mathbf{U}^\perp\|\,,$$

(7)

*then (3) holds and* $\mathcal{U}$ *is recovered exactly with probability at least* $1 - 5\delta$ *where* $\eta_\delta' = \max\left(\frac{4}{3}\log 2(r_o)/\delta\,,\ \sqrt{4\frac{n_o}{r_o}\log\frac{2r_d}{\delta}}\right)$ *and* $\xi = \frac{\min\left(\left\{\|\mathbf{b}_j^T\mathbf{U}^\perp\|_2\right\}_{j=1}^{n_o}\right)}{\|\mathbf{U}_o^T\mathbf{U}^\perp\|}$.

Theorem 3 indicates that $n_i/r$ should be sufficiently larger than $n_o/r_o$. If $r_o$ is comparable to $r$, it is in fact a necessary condition because we can not label the columns of $\mathbf{B}$ as outliers if $n_o$ is also comparable with $n_i$. If $r_o$ is large, the sufficient condition is similar to the sufficient conditions of Theorem 1 in which the outliers are distributed randomly on $\mathbb{S}^{M_1-1}$.

It is informative to compare the requirements of iSearch with the requirements of CoP. With iSearch, $n_i/r$ should be sufficiently larger than $\frac{n_o}{r_o}\|\mathbf{U}_o\mathbf{U}^\perp\|$ to guarantee that the algorithm distinguishes the outliers successfully. With CoP, $n_i/r_i$ should be sufficiently larger than $n_o/r_o + \|\mathbf{U}_o^T\mathbf{U}\|n_i/r_i$ [9,27]. The reason that CoP requires a stronger condition is that iSearch finds a direction for each outlier which is highly incoherent with $\mathcal{U}$.

## 4.4 Outlier Detection When the Inliers are Clustered

In the analysis of the robust PCA methods, mostly it is assumed that the inliers are randomly distributed in $\mathcal{U}$. In practise the inliers form several clusters in the column-space of the data. In this section, it is assumed that the inliers form $m$ clusters. The following assumption specifies the presumed model and Theorems 4 provides the sufficient conditions.

**Assumption 4.** *The matrix of inliers can be written as* $\mathbf{A} = [\mathbf{A}_1 \ ... \ \mathbf{A}_m]\mathbf{T}_A$ *where* $\mathbf{A}_k \in \mathbb{R}^{M_1 \times n_{ik}}$, $\sum_{k=1}^m n_{ik} = n_i$, *and* $\mathbf{T}_A$ *is an arbitrary permutation matrix. The columns of* $\mathbf{A}_k$ *are drawn uniformly at random from the intersection of subspace* $\mathcal{U}_k$ *and* $\mathbb{S}^{M_1-1}$ *where* $\mathcal{U}_k$ *is a d-dimensional subspace. In other word, the columns of* $\mathbf{A}$ *lie in a union of subspaces* $\{\mathcal{U}_k\}_{k=1}^m$ *and* $(\mathcal{U}_1 \oplus \ ... \ \oplus \mathcal{U}_m) = \mathcal{U}$ *where* $\oplus$ *denotes the direct sum operator.*

**Theorem 4.** *Suppose the distribution of the outliers/inliers follows Assumptions 1 to 4. Further define* $\mathcal{A} = \rho \left( \sqrt{\frac{2}{\pi}} \frac{n_g}{\sqrt{d}} - 2\sqrt{n_g} - \sqrt{\frac{2n_g \log \frac{1}{\delta}}{r-1}} \right)$ *where* $g = \arg\min_k \inf_{\substack{\delta \in \mathcal{U}_k \\ \|\delta\|=1}} \|\delta^T \mathbf{A}_k\|_1$, *and* $\rho = \inf_{\substack{\delta \in \mathcal{U} \\ \|\delta\|=1}} \sum_{k=1}^m \|\delta^T \mathcal{U}_k\|_2$ . *If the sufficient conditions in (4) are satisfied, then (3) holds and* $\mathcal{U}$ *is recovered exactly with probability at least* $1 - 7\delta$.

Since the dimensions of the subspaces $\{\mathcal{U}_k\}_{k=1}^m$ are equal and the distribution of the inliers inside these subspace are similar, roughly $g = \arg\min_k n_{ik}$ [19]. Thus, the sufficient conditions indicate that the population of the smallest cluster scaled by $1/\sqrt{d}$ should be sufficiently larger than $n_o/M_1$. The parameter $\rho = \inf_{\substack{\delta \in \mathcal{U} \\ \|\delta\|=1}} \sum_{k=1}^m \|\delta^T \mathcal{U}_k\|_2$ is similar to the permeance statistic introduced in [19]. It shows how well the inliers are distributed in $\mathcal{U}$. Evidently, if the inliers populate all the directions inside $\mathcal{U}$, a subspace recovery algorithm is more likely to recover $\mathcal{U}$ correctly. However, having a large value of permeance statistic is not a necessary condition. The reason that permeance statistic appears in the sufficient conditions is that we establish the sufficient conditions to guarantee the performance of iSearch in the worst case scenarios. In fact, if the inliers are close to each other or the subspaces $\{\mathcal{U}_i\}_{i=1}^m$ are close to each other, generally the performance of iSearch improves. The reason is that the more inliers are close to each other, the smaller their Innovation Values are.

## 5 Numerical Experiments

A set of experiments with synthetic data and real data are presented to study the performance and the properties of the iSearch algorithm. In the presented experiments, iSearch is compared with the existing methods including FMS [17], GMS [39], CoP [27], OP [36], and R1-PCA [6].

### 5.1 Phase Transition

In this experiment, the phase transition of iSearch is studied. Define $\hat{\mathbf{U}}$ as an orthonormal basis for the recovered subspace. A trial is considered successful if

$$\frac{\|(\mathbf{I} - \mathbf{U}\mathbf{U}^T)\hat{\mathbf{U}}\|_F}{\|\mathbf{U}\|_F} < 10^{-2} \ .$$

The data follows Assumption 1 with $r = 4$ and $M_1 = 100$. The left plot of Figure 2 shows the phase transition of iSearch versus $n_i/r$ and $n_o/M_1$. White indicates correct subspace recovery and black designates incorrect recovery. Theorem 1 indicated that if $n_i/r$ is sufficiently large, iSearch yields exact recovery even if $n_o$ is larger than $n_i$. This experiment confirms the theoretical result. According to Figure 2, even when $n_o = 3000$, 40 inliers are enough to guarantee exact subspace recovery.

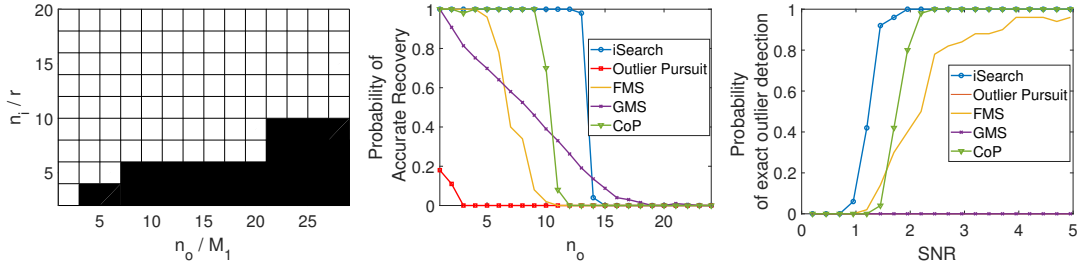

Figure 2: **Left panel:** The phase transition of iSearch in presence of the unstructured outliers versus $n_i/r$ and $n_o/M_1$ ($M_1 = 100$ and $r = 4$). **Middle panel:** The probability of accurate subspace recovery versus the number of structured outliers ($n_i = 100$, $\eta = 0.1$, $M_1 = 100$, and $r = 10$). **Right panel:** The probability of exact outlier detection versus SNR. The data contains 10 structured outliers and 300 unstructured outliers ($n_i = 100$, $n_o = 310$, $r = 5$, and $M_1 = 100$).

## 5.2 Structured Outliers

In this experiment, we consider structured outliers. The distribution of the outliers follows Assumption 2 with $\eta = 0.1$ and $M_1 = 100$. In addition, the inliers are clustered and they lie in a union of 5 2-dimensional linear subspaces. There are 20 data points in each subspace (i.e., $n_i = 100$) and $r = 10$. A successful trial is defined similar to Section 5.1. We are interested in investigating the performance of iSearch in identifying structured outliers when they are close to $\mathcal{U}$. Therefore, we generate vector $\mathbf{q}$, the center of the cluster of the outliers, close to $\mathcal{U}$. Vector $\mathbf{q}$ is constructed as $\mathbf{q} = \frac{[\mathbf{U}\ \mathbf{p}]\mathbf{h}}{\|[\mathbf{U}\ \mathbf{p}]\mathbf{h}\|_2}$, where the unit $\ell_2$-norm vector $\mathbf{p} \in \mathbb{R}^{M_1 \times 1}$ is generated as a random direction on $\mathbb{S}^{M_1-1}$ and the elements of $\mathbf{h} \in \mathbb{R}^{(r+1)\times 1}$ are sampled independently from $\mathcal{N}(0, 1)$. The generated vector $\mathbf{q}$ is close to $\mathcal{U}$ with high probability because the column-space of $[\mathbf{U}\ \mathbf{p}]$ is close to the column-space of $\mathbf{U}$. The middle plot of Figure 2 shows the probability of accurate subspace recovery versus the number of outliers. The number of evaluation runs was 50. One can observe that in contrast to the unstructured outliers, the robust PCA methods tolerate few number of structured outliers.

## 5.3 Noisy Data

In this section, we consider the simultaneous presence of noise, the structured outliers and the unstructured outliers. In this experiment, $M_1 = 100$, $r = 5$, and $n_i = 100$. The data contains 300 unstructured and 10 structured outliers. The distribution of the structured outliers follow Assumption 2 with $\eta = 0.1$. The vector $\mathbf{q}$, the center of the cluster of the structured outliers, is generated as a random direction on $\mathbb{S}^{M_1-1}$. The generated data in this experiment can be expressed as $\mathbf{D} = [\mathbf{B}\ \mathbf{A}_n]$. The matrix $\mathbf{A}_n = \mathbf{A} + \zeta\mathbf{N}$ where $\mathbf{N}$ represents the additive Gaussian noise, and $\zeta$ controls the power of the additive noise. Define SNR $= \frac{\|\mathbf{A}\|_F^2}{\|\zeta\mathbf{N}\|_F^2}$. Since the data is noisy, the algorithms can not achieve exact subspace recovery. Therefore, we examine the probability that an algorithm distinguishes all the outliers correctly. Define vector $\mathbf{f} \in \mathbb{R}^{M_2 \times 1}$ such that $\mathbf{f}(k) = \|(\mathbf{I} - \hat{\mathbf{U}}\hat{\mathbf{U}}^T)\mathbf{d}_k\|_2$. A trial is considered successful if

$$\max\Big(\{\mathbf{f}(k)\ :\ k > n_o\}\Big) < \min\Big(\{\mathbf{f}(k)\ :\ k \le n_o\}\Big).$$

The right plot of Figure 2 shows the probability of exact outlier detection versus SNR. It shows that iSearch robustly distinguishes the outliers in the strong presence of noise. The number of evaluation runs was 50.

## 5.4 Outlier Detection in Real Data

An application of the outlier detection methods is to identify the misclassified data points of a clustering method [9, 27]. In each identified cluster, the misclassified data points can be considered as outliers. In this experiment, we assume an imaginary clustering method whose clustering error is 25 %. The robust PCA method is applied to each cluster to find the misclassified data points. The clustering is re-evaluated after identifying the misclassified data points. We use the Hopkins155

dataset [33], which contains data matrices with 2 or 3 clusters. In this experiment, 27 matrices with 3 clusters are used (i.e., the columns of each data matrix lie in 3 clusters). The outliers are linearly dependent and they are very close to the span of the inliers since the clusters in the Hopkins155 dataset are close to each other. In addition, the inliers form a tight cluster. Evidently, the robust PCA methods which assume that the outliers are randomly distributed fail in this task. This experiment with real data contains most of the challenges that a robust PCA method can encounter. For more details about this experiment, we refer the reader to [9, 27].

Table 1 shows the average clustering error after applying the robust PCA methods to the output of the clustering method. One can observe that iSearch significantly outperforms the other methods. The main reason is that iSearch is robust against outliers which are closed to $\mathcal{U}$. In addition, the coherency between the inliers enhances the performance of iSearch.

Table 1: Clustering error after using the robust PCA methods to detect the misclassified data points.

| iSearch | CoP | FMS | R1-PCA | PCA |
|---|---|---|---|---|
| 2 % | 7 % | 20.3 % | 16.8 % | 12.1 % |

## 5.5    Activity Detection in Real Noisy Data

In this experiment, we use the robust PCA methods to identify a rare event in a video file. We use the Waving Tree video file [21]. In this video, a tree is smoothly waving and in the middle of the video a person crosses the frame. The frames which only contain the background (the tree and the environment) are inliers and the few frames corresponding to the event, the presence of the person, are the outliers. Since the tree is waving, the inliers are noisy and we use $r = 3$ for all the methods. In addition, we identify column $\mathbf{d}$ as outlier if $\|\mathbf{d} - \hat{\mathbf{U}}\hat{\mathbf{U}}\mathbf{d}\|_2 / \|\mathbf{d}\|_2 \geq 0.2$ where $\hat{\mathbf{U}}$ is the recovered subspace. In this experiments, the outliers are very similar to each other since the consecutive frames are quite similar to each other. We use iSearch, CoP, FMS, and R1-PCA to detect the outlying frames. iSearch, CoP, and FMS identified all the outlying frames correctly. R1-PCA could not identify those frames in which the person does not move. The reason is that those frames are exactly similar to each other. Figure 3 shows some of the outlying frames which is missed by R1-PCA.

+

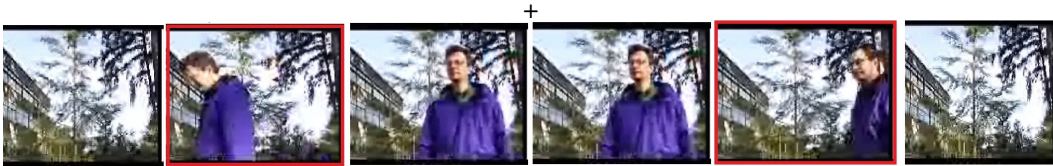

Figure 3: Some of the frames of the Waving Tree video file. The highlighted frames are detected as outliers by R1-PCA.

## 6    Conclusion

A new discovery about the applications of Innovation Search was presented. It was shown that the directions of innovation can be utilized to measure the innovation of the data points and to identify the outliers as the most innovative data points. In the robust PCA setting, the proposed approach recovers the span of the inliers using the least innovative data points. It was shown that iSearch can provably recover the span of the inliers with different models for the distribution of the outliers including randomly distributed outliers, linearly dependent outliers, and clustered outliers. In addition, analytical performance guarantees with clustered inliers were presented. The theoretical and numerical results showed that finding the optimal directions makes iSearch significantly robust to the outliers which carry weak innovation. Moreover, the experiments with real and synthetic data demonstrate the robustness of the proposed method against the strong presence of noise.

## Footnotes

[1] If the data is noisy, $r_d$ should be set equal to the number of dominant singular values. In this paper, we do not theoretically analyze iSearch in the presence of noise. In the numerical experiments, we set $r_d$ equal to the index of the largest singular value which is less than or equal to 0.01 % of the first singular value.

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
