[Reviews · NeurIPS 2019]

Reviewer 1



This paper defines a weighting scheme that allows for the isolation of outlier points that might bias principal subspace detection, even in the context of using Robust PCA estimators. The main idea is that when columns of the data matrix fall completely outside of the main subspace spanned the the other columns, this can be detected to allow for the "sparse" part of robust PCA to contain complete columns instead of completely random locations anywhere in the data matrix. The model itself (and to some extent the idea of weighting) is similar in spirit to this un-cited paper: - A.S. Charles, A. Ahmed, A. Joshi, S. Conover, C. Turnes, and M.A. Davenport, Cleaning up toxic waste: Removing nefarious contributions to recommendation systems. Proceedings of the ICASSP Vancouver, Canada, May 2013. however the authors provide some conditions on correctness for their algorithm, which is different than the re-weighted l1 scheme used in the citation above. Overall I thought this paper was good. One piece that was unclear to me was how the innovations coefficients in Figure 1 were so well behaved. There was a large amount of variation in the single vector (A^Tc) calculations in the left two panels of Figure 1, however the variation was not as large in the right panel. I did not see where any averaging over coefficients calculated using different c vectors was done, which I assume would be needed. A small comment in "Data Model 1": there is a comma missing in [B (A+N)]T --> should be [B, (A+N)]T. I would recommend that the authors change the name of the algorithm. iSearch is used a lot for different search engines, and a more unique name would help. Perhaps Innovations for Nixing Outliers (In N' Out)? Maybe that one isn't good, but the idea remains.

Reviewer 2



There is a lot of literature for this extensively studied problem; so no novelty on the problem setting. The authors present a new algorithm that is shown to be effective in challenging generating models where the outlier columns are very close to each other, or very close to the span of the inliers. In Data Model 1 definition, "m" is a typo. It should be M_1. The notation of [B (A+N)]T is not very clear, and potentially confusing, until you stare at the Notation section that comes later. This is an example of poor writing. The ideas in the iSearch algorithm seem to be reasonable but nothing strikes me as ground breaking. If inliers lie in a low dimensional linear subspace and we compute a direction that agrees maximally with any one such inlier, such a direction will be not so much in agreement with outliers that don't lie in this subspace. So it does seem like a relatively simple observation. The theoretical work is possibly what may make this paper acceptable to readers with a theoretical bent. That said, while the theorems seem to be analytical guarantees, it is difficult to see what that means practically. If the guarantees kick in only in conditions that lead to easy instances, then the usefulness of such theoretical guarantees is limited. So I'm mildly positive on the paper and recommend the below score.

Reviewer 3



This work presents a provable algorithm for detecting outliers based on robust PCA. I am impressed by the analysis and results on the different distributions of the outliers. The theoretical proof is solid. I only have a few concerns: a) In the introduction, the authors failed to mention many other robust PCA works. b) Can the authors introduce more on Assumption 1 and Assumption 2? Why are they valid and how are they related to practical applications?

[Author Response · NeurIPS 2019]

We would like to thank all the reviewers for their positive feedback and their helpful comments.

• **Common comment.** $[\mathbf{B}\ (\mathbf{A} + \mathbf{N})]$: We will define concatenation of matrices in Data model 1 and will change it to $[\mathbf{B},\ (\mathbf{A} + \mathbf{N})]$. Thanks for the suggestion.

## Responses to Reviewer 1

• *The ICASSP paper mentioned by the reviewer*: We thank the reviewer for bringing this paper into our attention. We will cite it along with the other matrix decomposition based methods ([1,3,32]). As the reviewer correctly indicated, our approach is different from the low rank plus sparse matrix decomposition based methods. In contrast to those methods, we do not need to assume that the number of outliers is significantly less than the number of inliers. In addition, the proposed optimization problem does not perform matrix decomposition. It is used to compute the innovation values.

• *Figure 1 and the variations:* The left plots shows the absolute inner product value between the optimal direction and all the data points while the right plot shows the innovation value for each data point. Each innovation value is computed using the average of $M_2$ absolute inner product values. Thus, the innovation values exhibit less variations.

• *Name of the algorithm:* As per reviewer's suggestion, we are considering alternative names such as Outlier Pursuit using Innovation Search (OPiS) or Robust PCA via Innovation Search (iSearch-PCA).

• *More experiments and data-sets:* In the paper, we specifically used the Hopkins155 data-set and the video file to exhibit the robustness of proposed method against structured outliers and outliers which are close to the span of inliers. In the revision, we will cite an extended version which contains further experiments.

## Responses to Reviewer 2

• *The idea of iSearch ("If inliers ... it does seem like a relatively simple observation"):* If the data point is an inlier, mostly its corresponding direction of innovation is not much different from the data point itself. The proposed approach shows its significance when we study the optimal direction corresponding to an outlier. In contrast to the inliers, when the data point is an outlier, the projection of its direction of innovation on the inliers can be significantly different from the projection of the data point itself on the inliers. That is the main reason the proposed method outperforms CoP and the other robust PCA methods on most of the challenging experiments because the optimal direction corresponding to an outlier is orthogonal or nearly orthogonal to the inliers.

• *The significance of the theoretical results and their practicality:* The presented results guarantee that the proposed approach can handle different types of outliers. They show that in contrast to some of the existing works, the proposed method is not limited to the unstructured outliers. Moreover, the theoretical results shed light on interesting features of the algorithm. For instance, Theorem 1 suggests that if the rank of $\mathbf{A}$ is sufficiently smaller than the dimension of the ambient space, the proposed approach can successfully recover the correct subspace even if the outliers dominate the data. This feature might sound counterintuitive but it is correctly predicted by the theorem. As another example, Theorem 4 shows that when the inliers are clustered, the population of the smallest cluster is the key factor (not necessarily the population of all of the inliers). Accordingly, the theoretical results not only serve as guarantees for the performance of the algorithm, but also they help to have a deeper understanding of the important features of the algorithms. In addition, if we compare the sufficient conditions of the proposed approach with the existing methods, the strengths of iSearch can be perceived. For instance, if we compare the requirements of the proposed approach with linearly dependent outliers against the corresponding sufficient conditions of CoP [26, 8], it can be observed that iSearch's sufficient conditions are notably simpler and less restrictive (discussion after Theorem 3). The reason is that the direction of innovation of an outlier is highly incoherent with the inliers (even if the outlier is coherent with them).

There is a short discussion after each theorem which discusses the important aspects of each result. As per reviewer's suggestion, we will extend these discussions in the revised paper to further clarify the significance of the results.

## Responses to Reviewer 3

• *Mentioning other works in Introduction*: In Introduction, we cite several works for both data corruption models ([1,3,32,35,16,7,19,4,6,23, 10,34,33,22]). As per reviewer's suggestion, we will provide a short description of the cited works and we will refer the reader to the section of related works for further discussion.

• *Explaining Assumption 1&2 and their practicality*: In most of the robust PCA papers, Assumption 1 is used to analyze the algorithm. In this assumption, the outliers are randomly distributed on the unit sphere. Assumption 1 can represent the scenarios in which the outliers are corresponding the data points which are overwhelmed with strong noise. However, this model can not represent the outliers in many other applications in which the outliers are structured. Accordingly, we introduced Assumption 2 and Assumption 3 to let the outliers to be structured. Assumption 2 let the outliers to form a cluster outside of the span of the inliers. This assumption is valid in the applications in which there are a few outliers which form a structure different from the structure of the inliers. For instance, in the activity detection example, the outliers are very similar to each other and they form a cluster. Assumption 3 let the outliers to be linearly dependent. Some of the existing methods make this restrictive assumption that a small subset of the outliers are not linearly dependent. However, in some applications (such as the experiment with Hopkins155 data set) the outliers are linearly dependent or there are repetitive outliers.

[Meta-Review · NeurIPS 2019]

This paper presents a new convex approach to robust PCA that is different from previously proposed convex approaches, and is shown to have good theoretical performance guarantees under several different outlier distributions. The paper seems careful and a new contribution (inspire of the method, like other convex approaches, being somewhat high-complexity). the reviewers are also in agreement that this should be accepted.